# In Silico Neuroprotective Effects of Specific *Rheum palmatum* Metabolites on Parkinson’s Disease Targets

**DOI:** 10.3390/ijms241813929

**Published:** 2023-09-11

**Authors:** Patrick Jay B. Garcia, Steven Kuan-Hua Huang, Kathlia A. De Castro-Cruz, Rhoda B. Leron, Po-Wei Tsai

**Affiliations:** 1School of Chemical, Biological, and Materials Engineering and Sciences, Mapúa University, Manila 1002, Philippines; pjbgarcia@mymail.mapua.edu.ph (P.J.B.G.); kadecastro@mapua.edu.ph (K.A.D.C.-C.); rbleron@mapua.edu.ph (R.B.L.); 2School of Graduate Studies, Mapúa University, Manila 1002, Philippines; 3Department of Medical Science Industries, College of Health Sciences, Chang Jung Christian University, Tainan 711, Taiwan; 7224837@mail.cjcu.edu.tw; 4Division of Urology, Department of Surgery, Chi Mei Medical Center, Tainan 711, Taiwan; 5School of Medicine, College of Medicine, Kaohsiung Medical University, Kaohsiung 807, Taiwan

**Keywords:** *Rheum palmatum*, Parkinson’s disease, neuroprotection, in silico, molecular docking, QSAR, ADMET properties

## Abstract

Parkinson’s disease (PD) is one of the large-scale health issues detrimental to human quality of life, and current treatments are only focused on neuroprotection and easing symptoms. This study evaluated in silico binding activity and estimated the stability of major metabolites in the roots of *R. palmatum* (RP) with main protein targets in Parkinson’s disease and their ADMET properties. The major metabolites of RP were subjected to molecular docking and QSAR with α-synuclein, monoamine oxidase isoform B, catechol *o*-methyltransferase, and A_2A_ adenosine receptor. From this, emodin had the greatest binding activity with Parkinson’s disease targets. The chemical stability of the selected compounds was estimated using density functional theory analyses. The docked compounds showed good stability for inhibitory action compared to dopamine and levodopa. According to their structure–activity relationship, aloe-emodin, chrysophanol, emodin, and rhein exhibited good inhibitory activity to specific targets. Finally, mediocre pharmacokinetic properties were observed due to unexceptional blood–brain barrier penetration and safety profile. It was revealed that the major metabolites of RP may have good neuroprotective activity as an additional hit for PD drug development. Also, an association between redox-mediating and activities with PD-relevant protein targets was observed, potentially opening discussion on electrochemical mechanisms with biological functions.

## 1. Introduction

Parkinson’s disease (PD) is the second most prevalent neurodegenerative disease after Alzheimer’s disease [1,2]. It has been considered the world’s fastest-growing neurological disorder and an economic burden for older adults because of medication costs and productivity losses [3]. This disease is characterized by bradykinesia, resting tremors, muscle stiffness, speech and writing changes, postural instability, and a wide array of motor and non-motor symptoms [4,5,6]. While the exact pathogenesis of PD is still unclear, primary observations reveal that such progressive disease develops from the degradation of dopaminergic neurons in the substantia nigra pars compacta (SNpc) in the basal ganglia, and the development of intraneuronal proteinaceous cytoplasmic inclusions called Lewy bodies [7]. Approximately four of five PD patients are idiopathic; the others are presumed of genetic origins [4]. Unfortunately, current treatments for PD are for neuroprotection and symptomatic therapy due to a vague understanding of its exact pathogenesis [5,8].

Dopamine (a catecholamine) is a crucial target for treating symptoms of PD since it is the primary neurotransmitter for movement coordination and control (in the motor circuitry of the brain in the basal ganglia), as well as in the cognition and reward system [9]. The death of dopaminergic neurons in SNpc consequently decreases dopamine production responsible for the motor symptoms of PD, such as its effect on the degradation of the nigrostriatal pathway responsible for sensory stimuli and movement, causing bradykinesia [9,10]. Because of the altered motor circuitry in the basal ganglia, automatic postural adjustments are disrupted, causing postural instability and immobility [11]. Also, thalamic activity from basal ganglia dysfunction was found to be promoted due to decreased dopamine, which regulates the thalamocortical loop, and led to abnormal cortical oscillations that cause resting tremors [12,13]. While the death of dopaminergic neurons is vital in PD indications, no complete conclusions on neuronal death have been reached. However, prior studies hypothesized that neuroinflammation from oxidative stresses, metabolic alterations, mitochondrial dysfunction, the impairment of autophagy functions, and intestinal inflammation from gut dysbiosis affecting the enteric nervous system (ENS) are some of the fundamental mechanisms of neuronal death [5,14,15,16].

Some specified mechanisms of neuronal death are the loss of mitochondrial complex I functionality responsible for dopamine production and fragmented mitochondria from mutations of coiled-coil-helix-coiled-coil-helix domain containing 2 (CHCHD2) [17,18,19]. Also, dysregulations in the tryptophan (Trp)–kynurenine (KYN) metabolic system observed in PD patients potentially led to further neuroinflammation due to the immunomodulatory properties of the metabolites, as well as the accumulation of neuronal death-causing compounds [17,20]. Indeed, imbalances in the Trp-KYN pathway were suspected of causing neurodegeneration due to a heightened ratio of neurotoxic to neuroprotective metabolites, such as the reported increase in quinolinic acid (QUIN) in blood plasma and 3-hydroxykynurenine (3-HK) in the cerebrospinal fluid of PD patients, which was implicated to increased oxidative stress, excitotoxicity, and neuroinflammation where dopaminergic neurons were particularly vulnerable [17,21,22]. Since dopamine is a precursor of other catecholamines, specifically epinephrine and norepinephrine, other non-motor functions—such as peristalsis, heart rate, and blood pressure regulation—are diminished for PD patients [10,23,24,25]. 

Gnanaraj et al. [8] summarized the PD-relevant proteins of utmost concern for their symptomatic treatment and neuroprotection. One of PD’s first genetic links is the overexpression of *SNCA*, an α-synuclein (ASN) coding gene, causing the formation of Lewy bodies from ASN aggregations [26]. On the other hand, in the standard treatment of PD, monoamine oxidase isoform B (MAOB) and catechol *o*-methyltransferase (COMT) are most considered for their function in dopamine metabolism [27]. Due to the impact of dopamine on human movement and control, its excessive decrease promotes uncontrolled and nonrigid movements. At the same time, the antagonism of the A_2A_ adenosine receptor (A_2A_AR) reduces the wear-off effect of dopaminergic therapies [28]. This receptor modulates motor circuits through inhibition of the direct pathway (motion-induction) and promotion of the indirect pathway (motion-arrest) [28,29]. Current treatments of PD include, but are not limited to, levodopa and its combination drug (such as carbidopa); dopamine agonists (such as apomorphine, ropinirole, cabergoline); MAOB inhibitors (such as rasagiline and selegiline); COMT inhibitors (such as entacapone and tolcapone); anticholinergics; anti-inflammatory agents; and vitamins A, E, and C [5,30].

Natural products have always been an abundant and diverse source of hit and lead compounds with good pharmacokinetic properties, including for treating neurological diseases [31,32]. Interestingly, complex mixtures of these medicines could provide curative potential to PD because of its multiple targeted approach from several potentially bioactive compounds. In traditional Chinese medicine, the roots of *Rheum palmatum* (RP) or Chinese rhubarb are used for various therapeutic uses, including PD symptom treatment [33]. Previously, it has been revealed that anthraquinones (specifically chrysophanol, rhein, aloe-emodin, emodin, and physcion as shown in Figure 1) are abundant in RP [34]. Consequently, neurotransmitters such as dopamine, epinephrine, norepinephrine, and levodopa show redox-mediating activity for microbial fuel cells (MFCs) and quasi-reversible redox ability in cyclic voltammetry measurements [35]. Also, PD medications were observed as quasi-reversible redox mediators where the electrochemical mechanism for biological function is still open for follow-up investigations [36]. Since RP extracts show good electron-shuttling characteristics, the major metabolites of RP may have similar activity as the treatment mentioned [34]. Indeed, prior studies have explored that emodin and chrysophanol exhibited neuroprotective properties in vitro and in vivo for PD [37].

A molecular simulation study was conducted for RP’s major metabolites, revealing the potential alternative therapeutic compounds or conjunctive use of RP extracts with conventional medicines for PD. In silico investigations of major compounds of RP were performed, considering neurotransmitters, and PD medicines have good MFC bioenergy amplification. This investigation evaluated their potencies to relevant PD targets and safety profiles with their binding affinity, pharmacokinetic properties, expected reactivity, and quantitative structure–activity relationship for the prediction of their inhibitory action.

## 2. Results and Discussion

### 2.1. Molecular Docking

Primary metabolites from RP were subjected to the ligand preparation protocol in Discovery Studio, which generated the 31 ligands presented in Appendix A. Two standard drugs used for PD treatment (dopamine and levodopa) were used as positive controls and were prepared similarly to the mentioned metabolites. Validation of the protein structures of protein targets for PD revealed that the root-mean-square-difference (RMSD) of minimized structures compared to experimentally elucidated structures were no more than 2.0 Å, which indicated valid structures for docking studies (see Appendix A) [38,39]. Each ligand form was screened using hotspots-based docking, using the LibDock protocol available in Discovery Studio reported in Table 1. The ligand conformers for each metabolite with the highest LibDock score were selected for further analysis to CDOCKER, a CHARMm-based docking protocol.

Furthermore, 35 protein–ligand complexes were refined through CDOCKER with the top hits presented in Table 2, where CDOCKER energy combined protein–ligand binding interaction energy and ligand conformational energy. Compounds above or within the dopamine and levodopa standard range were analyzed for their molecular interaction and properties. While binding these metabolites rendered them inferior to levodopa, multiple ligands inhibited many targets. In this condition, emodin exhibited the best binding activity among all protein targets. Specifically, ASN (PDB ID: 1XQ8) interacted the best with emodin (−34.4961 kcal/mol), surpassing dopamine (−31.0394 kcal/mol). There are only functional assumptions of ASN on dopamine regulation from its ability for synaptic vessel provision in presynaptic terminals and cytoskeletal dynamics from microtubular interaction [40,41]. Interestingly, one symptomatic progression of PD concerns alternate folding of ASN forming insoluble fibrils, which further aggregates leading to neurodegeneration, and they are postulated as causative to prion-like cell-to-cell transmission of ASN misfolding [42]. Nevertheless, the aggregation of ASN is widespread in numerous brain regions of the central nervous system (CNS) for PD patients. These aggregations impair numerous neural functions, such as proteasomal and lysosomal functions, potentially hindering protein clearance and promoting neuronal death [26,43]. Additionally, there are preliminary investigations on ASN aggregation due to CHCHD2 mutations and interaction with 3-HK. Hence, it was suspected that interactions with ASN could inhibit the formation of Lewy bodies for decreased neurotoxicity.

Nevertheless, chrysophanol (−39.7554 kcal/mol) and physcion (−30.3292 kcal/mol) may have a good interaction with MAOB (PDB ID: 2C65). At the same time, the best-interacting compound for COMT (PDB ID: 3BWM) is emodin (−32.431 kcal/mol). Normally, MAOB catalyzes the oxidative deamination of xenobiotic and biogenic amines, including dopamine [44]. In contrast, COMT normally regulates levels of catecholamines in the brain and peripheral tissues, especially levodopa, by its inactivation through methyl group transfer from the S-adenosylmethionine to the catechol structure [27,44]. For this reason, their inhibition ameliorates the motor dysfunction and fluctuations observed in PD patients and offsets the deterioration of dopaminergic neurons. Also, emodin has superior binding activity (−57.5427 kcal/mol) to A_2A_AR (PDB ID: 3EML), comparable to levodopa (−61.8193 kcal/mol). At the same time, rhein (−39.4557 kcal/mol) has a similarity in binding affinity with dopamine (−39.3845 kcal/mol) to A_2A_AR (PDB ID: 3EML). Due to the inhibitory interaction of A_2A_AR with the dopamine D2 receptor, reduced availability of dopamine in specific periods exacerbates motor fluctuations [28,29]. Hence, inhibition of A_2A_AR lessens the worsening of PD symptoms, such as dyskinesia, making it a non-dopaminergic treatment, usually in combination with dopaminergic treatments [28].

Frequent drug–receptor interactions of known drugs, such as hydrophobic contact, hydrogen bonding (HB), and π-stacked interactions [45], were observed in the selected protein–ligand complexes shown in Figure 2. The ASN-emodin complex formed a salt bridge between LYS45 and the alkoxide of the *o*-dihydroxy aromatic moiety in emodin. Alkyl and π-alkyl interactions were also a contributor to the stability of this complex. For the MAOB-chrysophanol complex, an attractive charge between LYS296 and the alkoxide of the phenolic structure, as well as HB with SER59 and TYR60, caused stabilization of the formed complex. This complex is further stabilized with stacked π-π, π-alkyl, and alkyl interactions. The superior binding of the MAOB-emodin complex formed an attractive charge between LYS296 and the alkoxide and is further stabilized with π-sulfur, stacked π-π, π-alkyl, and alkyl interactions. At the same time, the MAOB-physcion complex formed HB between CYS172 and TYR435 with π-sulfur, stacked π-π, T-shaped π-π, π-alkyl, and alkyl interactions, suggesting excellent molecular orbital alignment. The COMT-emodin complex formed HB between ASP141 and ASN170 and an attractive charge further supported by an attractive charge, π-cation, π-sulfur, π-alkyl, and alkyl interactions. For the A_2A_AR-emodin complex, salt bridges and attractive charges (ARG107, ARG206, LYS1060, ARG1008), as well as HB (ARG107, ARG199, ARG206, and GLU1005), were formed; however, there was an unfavorable donor–donor interaction (ARG1008) which could potentially destabilize the complex structure. Finally, the A_2A_AR-rhein complex formed HB (ARG199, LEU202, ALA203, ARG206, and ARG1008), attractive charge (ARG107), and π-alkyl (LEU202 and ALA203). These interactions reveal considerable effects on the PD targets; however, this static simulation could not fully uncover the protein conformation changes that could affect protein–ligand interactions. Hence, further analysis with molecular dynamics simulations could uncover these details.

### 2.2. Density Functional Theory (DFT) Analysis

The energies of frontier molecular orbitals (FMO) can provide helpful quantum mechanical calculations describing compounds’ chemical reactivity and stability. According to Koopmans’ theorem, the energy of the highest occupied molecular orbital (ε_HOMO_) can be approximated to the required first ionization energy, and the energy of the lowest unoccupied molecular orbital (ε_LUMO_) approximates the electron affinity in the last occupied orbital. In contrast, the energy gap (Δε) relates to the reactivity of the compound [46,47,48,49]. Global reactivity descriptors from approximated ionization energies and electron affinity revealed beneficial molecular properties. In a chemical process, the hardness,
(1)$$\eta =\frac{1}{2}\left(\varepsilon_{\mathrm{LUMO}}-\varepsilon_{\mathrm{HOMO}}\right),$$
of molecules describes the resistance of electron cloud deformation to trivial perturbations. In contrast, the chemical potential,
(2)$$\mu =\frac{1}{2}\left(\varepsilon_{\mathrm{LUMO}}+\varepsilon_{\mathrm{HOMO}}\right),$$
shows the stability of the compounds through their electron transfer capability, where negative values indicate the compound for less propensity for decomposition [46,49,50]. The electrophilic index,
(3)$$\omega =\frac{\mu^{2}}{2\eta },$$
is one of the significant descriptors, since it describes potential affinity to a receptor for drug–receptor interactions. This descriptor assesses the inclination of the compound to acquire additional charge and its resistance to electronic charge exchange with the surroundings, indicative of electrophilicity [50]. A large energy band gap between the HOMO and LUMO is classified as hard molecules and is consequently less reactive. Conversely, soft molecules are more reactive for smaller energy band gaps [50,51]. From this, excellent inhibitors are characterized by their heightened stability.

Compared to the minimized structures (in Appendix A), all protein–ligand complexes except rhein in complex with A_2A_AR (PDB ID: 3EML) exhibited an increased energy band gap, revealing increased reactivity. Emodin (2.4536 eV) with ASN (PDB ID: 1XQ8) increased from 2.2949 eV. At the same time, chrysophanol (2.6025 eV), emodin (2.3333 eV), and physcion (3.0351 eV) with MAOB (PDB ID: 2C65) increased from 2.5902, 2.1354, and 2.9551 eV. At the same time, emodin (2.3313 eV and 2.3155 eV) with COMT (PDB ID: 3BWM) and A_2A_AR (PDB ID: 3EML) increased from 2.1354 eV. However, rhein (2.7222 eV) with A_2A_AR (PDB ID: 3EML) indicated a decrease from 2.8997 eV. Compared to the standards, MAOB-physcion and A_2A_AR-rhein exhibited better protein–ligand interaction than levodopa. Similarly, ASN-emodin and MAOB-chrysophanol have potentially more stable protein–ligand interactions. The chemical hardness of the ligands shown in Table 3 indicated less polarizability, where the mentioned are again superior in stability. Moreover, chemical potentials reveal chemical stability from decomposition and electron escaping tendency. All ligands were considered to have superior stability compared to the standards. The electrophilicity indices show rhein and physcion as least nucleophilic.

### 2.3. Predicted Inhibitory Activity from 3D QSAR Models

Quantitative structure–activity relationship models were established in this study for the prediction of the inhibitory activity of RP major metabolites on PD targets. Steric (from van der Waals potential of carbon atom probe) and electrostatic (from H^+^ point charge probe) contributions were considered in the 3D grid space of aligned and minimized ligands with known half-maximal inhibitory concentrations. A dataset of 45 known ASN inhibitors, 140 known MAOB inhibitors, 105 known COMT inhibitors, and 120 known A_2A_AR inhibitors was considered in the 3D QSAR model building. As shown in Table 4, all models are considered robust and reliable from set significance based on their internal validity. For a further assessment of their accuracy, cross-validation of the models reveals the most accuracy from MAOB (0.488) > A_2A_AR (0.400) > COMT (0.303) > ASN (0.225). On the other hand, external validation reveals the most accuracy from MAOB (0.458) > COMT (0.433) > ASN (0.387) > A_2A_AR (0.175). Appendix A indicate predicted and experimental pIC_50_ values in the training and test datasets, including their residuals.

All predicted values of IC_50_ to each protein target are within the dataset range as shown in Table 5. Consistently, emodin (15.957 µM) and rhein (18.457 µM) had the best inhibition of ASN among all metabolites. Compared to epigallocatechin-3-gallate with fibrillogenic inhibitory activity (9.8 µM), the best RP metabolites had 1.63 and 1.88-fold less inhibitory activity [52]. This inhibitory activity on ASN could decrease fibril formation and aggregation, thereby decreasing the toxicity of the oligomeric Lewy bodies to the neurons. Similarly, chrysophanol (0.088 µM) and emodin (0.128 µM) had similar outcomes as their binding affinity with MAOB, which are comparable with selegiline (0.0013 µM), a known MAOB inhibitor [53]. For this, the inhibition of MAOB slows down dopamine degradation, which restores the motor functions of PD patients. Compared to entacapone (0.386 µM) for COMT inhibition [54] and istradefylline (5.25 µM) for A_2A_AR antagonism [55], the comparable inhibition from aloe-emodin (0.330 µM) and good inhibition with A_2A_AR of 1.503 µM was revealed, based on its structural features. This interaction could potentially decrease the metabolic activity of levodopa for its increased therapeutic effect and sustained dopamine levels. Also, the decreased blocking activity of A_2A_ receptors to dopamine D2 receptors could potentially restore the balance of the motor circuitry. Nevertheless, chrysophanol (0.816 µM) was shown to have good inhibitory activity with A_2A_AR. Previous literature experimentally measured the inhibitory concentrations of the major metabolite on MAOB, where only emodin was observed to have appreciable inhibitory activity [56]. However, these results are open to further validations, specifically from immunochemical assays, protein–ligand binding assays, mutagenesis studies, functional assays, and X-ray crystallographic analyses.

### 2.4. ADMET Properties

Predicted pharmacokinetics and toxicity profiles from structure–activity relationships were performed in ADMETboost and pkCSM, as summarized in Table 6. All major metabolites of RP appeared to satisfy Lipinski’s rule of five. Only chrysophanol did not follow the Pfizer rule, which could potentially have a toxic profile. At the same time, both chrysophanol and physcion do not follow the GSK rule, which may provide subpar ADMET properties. Nevertheless, the physicochemical properties of these metabolites follow drug-like soft rules related to their ease of metabolism without substantially producing bioactive metabolites or intermediates [57]. For this reason, undesired or toxic side effects from metabolic products and drug–drug interactions are minimized. All compounds have desirable n-octanol/water partition coefficients; however, the metabolites are significantly hydrophobic compared to the standards.

Chrysophanol and physcion had good Caco-2 permeability with more than −5.15 log cm/s compared to the standards. On the other hand, all major RP metabolites, including the standards, have excellent human intestinal absorption (with more than 70% probable), which indicates a decreased requirement for parenteral administration. Since traditional medicines commonly use the oral route of administration and aqueous solvent, the good n-octanol/water distribution (1–3 log mol/L) observed for all metabolites and standards and poor aqueous solubility (−4–0.5 log mol/L) could potentially lead to decreased potency of RP but with optimum lipophilicity preferred for medicines for optimal lipid bilayer penetration [59,60]. Nevertheless, the oral bioavailability of the metabolites and standards are comparable with all compounds possessing a good plasma protein binding rate, since they have less than a 90% probability of combining completely with blood serum proteins. This result means that major RP metabolites could potentially have quick onset action, and they can reach the target site rapidly [61,62]. Chrysophanol, physcion, and rhein may cross the blood–brain barrier lightly (between 30–70%) compared to other metabolites and better than levodopa, potentially not requiring advanced drug delivery techniques for therapeutic effect. For these reasons, the metabolites may have good absorption and distribution to affect the required targets appreciably.

The compounds were found to have mediocre CYP2C9 and CYP2D6 inhibitory activities. However, all compounds were revealed to have an excellent probability of CYP2D6 inhibition. The CYP2D6 enzyme has mediocre metabolic activity in these compounds. This enzyme is highly expressed in the SN region of the CNS and is discouraged as a main metabolic route for CNS drugs due to their polymorphic nature [63]. While the half-life of the metabolites was superior to the standards, corresponding to a lengthier action, the hepatocyte clearance was faster than the standards, and the microsome clearance was slower. For this, a cumulative effect may be observed, potentially leading to accumulative ability, and their lipophilicity could have a toxic effect. At the same time, the metabolites reveal a toxicity profile akin to the standards. There is an observed small probability for hERG blocking potential, which could lead to cardiotoxicity. However, this probability could be assumed negligible since the RP metabolites were approximately like the standards. Further investigations into the interaction of the metabolites to hERG with immunochemical assays, molecular simulations, and electrophysiological studies could validate the predicted cardiotoxicity profile. However, mediocre mutagenicity and hepatotoxicity were observed for all metabolites. On the other hand, aloe-emodin, chrysophanol, and physcion were observed to have significantly higher tolerable doses than the other metabolites and the standards, and all metabolites had significantly low lethal doses for oral application. In fact, in vitro and in vivo investigations showed that excessive use of emodin manifested hepatotoxicity, renal toxicity, and reproductive toxicity with chrysophanol [64]. For this reason, the dosage of these compounds requires careful consideration since accumulation could be lethal.

All in silico pharmacokinetics and toxicity property results were consistent with other tools like pkCSM and ADMETlab 2.0. These findings indicate that RP metabolites may have a favorable therapeutic effect as a neuroprotective agent.

## 3. Materials and Methods

### 3.1. Preparation of Ligands

This study employed the molecular modeling and visualization software BIOVIA Discovery Studio 2022 (Dassault Systèmes, Vélizy-Villacoublay, France) for molecular simulations. Two-dimensional (2D) structure-data files (SDFs) of the identified ligand molecules were collected from the PubChem database (chem.ncbi.nlm.nih.gov, accessed on 10 May 2023) and prepared with the Prepare Ligands protocol in Discovery Studio (see Appendix A). Numerous ionization states were generated under pH 7.5 ± 1.0. A structure visualization of the ligands was generated from ChemDraw v.19.

### 3.2. Preparation of Proteins

Experimentally elucidated structures of protein targets available from the Protein Data Bank (PDB, www.rcsb.org, accessed on 10 May 2023) in May 2023 were collected in PDB file format. Afterwards, water molecules and irrelevant heteroatoms for the simulation were removed, and polar hydrogens were added to the protein structure. Minimization of the protein structure with co-crystal ligands with the CHARMm forcefield was performed to verify the accuracy of the results. The protein was prepared with the Prepare Protein protocol in Discovery Studio, which automatically repaired and protonated the protein structure under a pH of 7.4 and 0.145 M of ionic strength with 0.9 kcal/mol cutoff energy. Missing loops were built in the protocol using PDB SEQRES loop definition, limited to 20 residues, with loop refinement through CHARMm forcefield minimization.

### 3.3. Molecular Docking

The initial docking runs were performed using the LibDock algorithm in Discovery Studio to screen ligand forms. Numerous conformations were generated with conjugate-gradient and quasi-newton minimization with an energy threshold between isomers set to 20.0 kcal/mol. A maximum fraction of steric clashes of 10% and a final clustering radius of 0.5 Å were set for the docking. The fractions of acceptable solvent accessible surface area (SASA) cutoffs were set to 15% for apolar SASA and 5% for polar SASA with 18 grid points. The binding sites were determined from the literature and PDB site records (see Appendix A) [8,65,66,67,68,69]. Docking optimization was performed using a grid-based CDOCKER algorithm in Discovery Studio for the evaluation of the best ligand for each protein target. Random conformations were generated from 1000 dynamics steps at 1000 K target temperature or with consideration of electrostatic interactions. The docked conformations were refined with 800 maximum bad orientations and a vdW energy threshold of 300 kcal/mol. This complex was annealed for 2000 heating steps until 700 K and subsequently cooled for 5000 steps to 300 K. Final minimizations of the refined ligand pose were set to full potential. The docking analysis results were further evaluated through a comparison to standard drugs of Parkinson’s disease, i.e., dopamine and levodopa [5,30].

### 3.4. 3D QSAR Modeling and Activity Prediction

The QSAR model for each protein target was modeled after IC_50_ values of known interacting compounds collected in the ChEMBL database (ebi.ac.uk/chembl, accessed on 10 May 2023) and enumerated in Appendix A. These compounds were prepared using the Prepare Ligands for QSAR protocol in Discovery Studio, where duplicate data were removed, compound structure and charge were repaired, and numerous ionization states were generated under pH 7.4. The compounds were subjected to CHARMm minimization with Momany–Rone ligand partial charge estimation until no energy gradient was reached. Outlier compounds were removed through software-predefined parameters (atomic logarithmic 1-octanol/water partition coefficient, molecular weight, number of H donors, H acceptors, rotatable bonds, rings, aromatic rings, and fractional polar surface area). Outliers were classified under Poisson distribution considering short-ranged parameters with a *p*-value of 0.01. Rigid molecular overlay for compound alignment was performed under consensus-based, 50% steric, and 50% electrostatic field fit. The IC_50_ values of the compound data were converted into pIC_50_ values. These data are separated by diverse split through the predefined parameters into 80% training set and others for the test set. Three-dimensional QSAR models based on shape (steric) and charge (electrostatic) complementarity were generated for each protein target with pIC_50_ values as the activity variable, grid spacing of 1.5 Å, and 5-fold cross-validation. The 3D QSAR model correlates the relationship between the activity variable and molecular fields following Equation (4):(4)$$A=\sum_{i=1}^{n_{EP}}k_{EP,i}V_{EP,i}+\sum_{i=1}^{n_{vdW}}k_{vdW,i}V_{vdW,i}$$
Variables in this model are *n*, the number of descriptors which are each grid point; *k*, the model coefficient for the descriptor; and, *V*, the descriptor value at the grid point corresponding to the compound. These models are built with partial least squares (PLS) regression.

### 3.5. Density Functional Theory (DFT) Analysis

Molecular properties of ligand structures in the form of frontier molecular orbitals, band gap (Δε), and electrostatic potential maps were evaluated with the DMol^3^ package in Discovery Studio for the structure stability analysis. Reference ligands were optimized with CHARMm forcefield minimization. Single-point energy calculation of the reference and docked ligands was performed with Becke’s three-parameter, Lee–Yang–Parr (B3LYP) exchange-correlation functional and aqueous environment using the COSMO continuum solvent models. A double-numeric quality basis set with polarization functions (DNP) and a self-consistent field (SCF) density convergence of 1.0 × 10^−6^ was used for the analysis.

### 3.6. ADMET Properties Prediction

SMILES notation records of the prepared ligands were collected from the PubChem database (chem.ncbi.nlm.nih.gov, accessed on 10 May 2023). ADMETboost (ai-druglab.smu.edu/admet, accessed on 10 May 2023) was used for ADMET properties prediction for further analysis. This predictive model utilized MACCS, extended connectivity circular, Mol2Vec, and PubChem fingerprints, as well as Mordred and RDKit descriptors through the extreme gradient boosting machine learning modeling model [58]. Parameters include blood–brain barrier (BBB) permeability, metabolism (inhibition and substrate activity on human cytochrome P450 family), half-life, human ether-a-go-go-related gene (hERG) for cardiotoxicity, human hepatotoxicity, and AMES test for mutagenicity. The human maximum tolerable dose was determined from pkCSM (biosig.lab.uq.edu.au/pkcsm, accessed on 10 May 2023) [70]. Data visualization was generated from GraphPad Prism v.9.5.0. 

## 4. Conclusions

With increased patient cases, limited treatment diversity, and the complex and unclear pathogenesis of PD, the current state of research is far beyond a firm and near-complete understanding of the disease and, consequently, remains complicated with regards drug development. Current efforts for PD treatment continue to be symptomatic treatment and neuroprotection to delay its progression. Traditional medicines have been an excellent source of diverse compounds, and because of the complex mechanisms involved in PD, a multiple-targeted approach could potentially uncover interesting mechanisms and interactions. This study initially addressed the interaction of major RP metabolites for neuroprotective activities to Parkinson’s disease in silico, and emodin could be attributed as the main effect compound in the extracts. These metabolites inhibited Parkinson’s disease targets on par with controls and known inhibitors from its binding activity, and predicted the inhibition activity from structure–activity relationships with good chemical stability. 

## 5. Future Directions

While there is an interesting outlook on the multi-targeted effect of RP metabolites on key PD targets, experimental analyses are required to validate the computational results from this study. A more thorough computational analysis (such as network pharmacology and molecular dynamics simulations) could be initially performed for the prediction of the mechanism of action. In addition to previously mentioned validation studies, the activity of RP extracts or decoctions, including such extracts with in vitro and in vivo models of PD, may provide additional insights into its treatment. Gene expression studies and protein–ligand interaction experiments are recommended for further interpretation of their mechanism of action. Following this, structure isolation and comparative explorations of these extracts should provide the structural moiety and molecular interactions necessary for PD treatment. Before advanced investigation and clinical trials, these compounds should be subjected to experimental pharmacokinetics and toxicity studies.

## Figures and Tables

**Figure 1 ijms-24-13929-f001:**
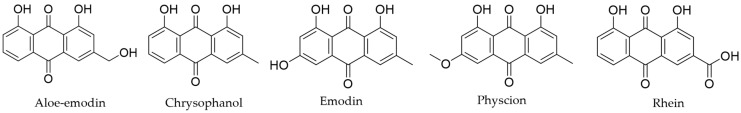
Chemical structures of RP major metabolites.

**Figure 2 ijms-24-13929-f002:**
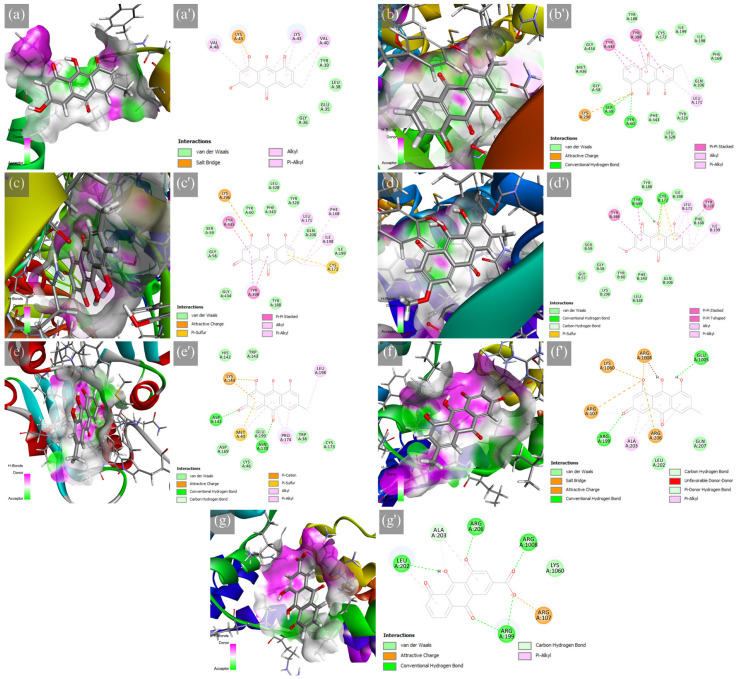
Hydrogen bond surface of protein receptor and 2D molecular interactions of major RP metabolites to PD targets, which are the (**a**,**a′**) ASN-emodin, (**b**,**b′**) MAOB-chrysophanol, (**c**,**c′**) MAOB-emodin, (**d**,**d′**) MAOB-physcion, (**e**,**e′**) COMT-emodin, (**f**,**f′**) A_2A_AR-emodin, and (**g**,**g′**) A_2A_AR-rhein.

**Table 1 ijms-24-13929-t001:** LibDock scores of RP major metabolites to PD targets.

Ligand	Form No.	ASN (PDB: 1XQ8)	MAOB (PDB: 2C65)	COMT (PDB: 3BWM)	A_2A_AR (PDB: 3EML)
Aloe-emodin	1	83.2529	121.666	102.147	103.938
	2	88.0637	125.89	122.628	119.698
	3	86.1153	122.002	108.446	106.68
	4	81.5402	121.551	101.605	103.695
	5	85.6552	121.304	115.675	107.195
	6	81.7519	122.844	101.408	104.847
Chrysophanol	1	78.9452	114.483	92.5595	98.8463
	2	79.5134	115.194	102.129	101.463
	3	77.4547	114.957	95.9588	100.238
	4	72.7176	115.208	93.4339	96.1831
	5	72.0571	114.407	93.139	98.3591
	6	64.9091	115.13	93.61	96.686
Emodin	1	81.1783	120.25	96.7097	97.0337
	2	84.4642	119.622	101.456	100.972
	3	83.4505	121.64	104.623	101.864
	4	82.0401	119.959	89.9023	95.9239
	5	83.5858	119.78	101.984	99.3968
Physcion	1	84.0608	124.603	98.9597	102.614
	2	88.5065	128.092	116.342	114.673
	3	86.15	128.415	118.163	105.413
	4	83.3815	125.267	97.6397	100.39
	5	88.3428	124.502	113.259	118.168
	6	83.8676	125.486	89.3911	99.4669
Rhein	1	86.5327	123.458	106.972	106.412
	2	84.0283	124.371	119.125	108.289
	3	87.7195	125.19	107.756	109.683
	4	84.7186	123.958	94.2238	106.914
	5	86.7861	124.038	98.3459	108.707
	6	82.2617	124.203	96.3977	106.131
Dopamine ^1^		65.7182	85.4606	89.5598	77.5231
Levodopa ^1^		75.4623	103.043	110.127	91.2493

^1^ Standard drugs.

**Table 2 ijms-24-13929-t002:** CDOCKER energy (in kcal/mol) of RP major metabolites to PD targets.

Ligand	ASN (PDB: 1XQ8)	MAOB (PDB: 2C65)	COMT (PDB: 3BWM)	A_2A_AR (PDB: 3EML)
Aloe-emodin	−10.9777	−3.33321	−12.2531	—
Chrysophanol	−9.65713	−39.7554	−10.606	−15.9783
Emodin	−34.4961	−44.9155	−32.431	−57.5427
Physcion	−10.5583	−30.3292	−15.7917	—
Rhein	−29.9042	−22.7501	−14.9567	−39.4557
Dopamine ^1^	−31.0394	−30.2512	−36.1948	−39.3845
Levodopa ^1^	−49.7879	−53.7193	−46.8662	−61.8193

^1^ Standard drugs.

**Table 3 ijms-24-13929-t003:** Electronic properties of standards and docked conformations of ligands (in eV).

Protein	Ligand	ε_HOMO_	ε_LUMO_	Δε	η	µ	ω
ASN	Emodin	−5.2390	−2.7854	2.4536	1.2268	−4.0122	6.5608
MAOB	Chrysophanol	−5.0907	−2.4882	2.6025	1.3013	−3.7895	5.5178
	Emodin	−5.1769	−2.8436	2.3333	1.1667	−4.0102	6.8924
	Physcion	−6.3604	−3.3253	3.0351	1.5175	−4.8429	7.7275
COMT	Emodin	−5.1784	−2.8472	2.3313	1.1656	−4.0128	6.9073
A_2A_AR	Emodin	−5.1753	−2.8598	2.3155	1.1578	−4.0175	6.9706
	Rhein	−6.0684	−3.3462	2.7222	1.3611	−4.7073	8.1400
	Dopamine ^1^	−5.1946	−2.7709	2.4237	1.2119	−3.9828	6.5447
	Levodopa ^1^	−4.5353	−1.8210	2.7143	1.3571	−3.1782	3.7213

^1^ Minimized structures of standard drugs.

**Table 4 ijms-24-13929-t004:** 3D QSAR model parameters.

Parameter	ASN	MAOB	COMT	A_2A_AR
N	36	112	84	96
pIC_50_ range	3.897–6.721	4.311–9.511	3.886–8.700	4.26–11.721
IC_50_ range (µM)	0.190–126.77	0.0003–48.87	0.002–130.02	1.9 × 10^−6^–75.86
**Internal validation**				
r	0.971	0.974	0.975	0.969
r^2^	0.942	0.948	0.951	0.939
r^2^_adj_	0.941	0.946	0.950	0.938
RMS residual error	0.183	0.249	0.243	0.245
**Cross-validation**				
q^2^	0.225	0.488	0.303	0.400
RMS residual error	0.674	0.781	0.927	0.772
**External validation**				
q^2^	0.387	0.458	0.433	0.175
RMS error	0.925	1.073	0.787	0.917
Mean absolute error	0.821	0.901	0.620	0.805

**Table 5 ijms-24-13929-t005:** Predicted IC_50_ (in µM) of standards and RP metabolites on PD targets.

Ligand	ASN	MAOB	COMT	A_2A_AR
Aloe-emodin	26.1921	0.2404	0.3297	1.5034
Chrysophanol	21.4724	0.0880	0.7712	0.8162
Emodin	15.9566	0.1277	0.7888	2.8790
Physcion	20.8521	1.2842	0.8107	23.0096
Rhein	18.4570	0.4140	0.7221	4.1237

**Table 6 ijms-24-13929-t006:** ADMET properties in silico with ADMETboost and pkCSM webserver.

	ALO	CHR	EMO	PHY	RHE	DOP	LDP
Physicochemical Properties							
Molecular weight	270.24	254.24	270.24	284.27	284.22	153.18	197.19
Hydrogen bond acceptors	5	4	5	5	5	3	4
Hydrogen bond donors	3	2	3	2	3	3	4
Rotational bonds	1	0	0	1	1	2	3
TPSA (Å^2^)	94.83	74.6	94.83	83.83	111.9	66.48	103.78
log K_O/W_	1.37	2.18	1.89	2.19	1.57	0.60	0.05
Lipinski rule	(+)	(+)	(+)	(+)	(+)	(+)	(+)
Pfizer rule	(+)	(−)	(+)	(+)	(+)	(+)	(+)
GSK rule	(+)	(−)	(+)	(−)	(+)	(+)	(+)
Golden triangle	(+)	(+)	(+)	(+)	(+)	(−)	(−)
**Absorption**							
C2P (log cm/s)	−5.30	−5.06	−5.25	−5.11	−5.37	−5.33	−5.34
HIA (%)	73.81	73.93	73.93	73.93	73.81	73.58	73.99
log D_7.4_	1.77	1.93	1.86	1.98	1.89	1.51	1.43
log S (log mol/L)	−4.79	−5.00	−4.79	−5.09	−4.72	−4.24	−4.35
Oral bioavailability (%)	42.60	42.67	41.87	45.12	43.18	41.13	48.56
**Distribution**							
BBB penetration (%)	29.99	33.86	29.47	31.65	33.29	29.16	30.49
PPBR (%)	42.67	48.22	45.28	45.46	54.58	39.48	50.56
**Metabolism**							
CYP2C9 inhibitor (%)	56.55	58.07	63.77	58.02	54.56	47.99	47.19
CYP2C9 substrate * (%)	35.14	34.41	34.33	36.50	37.90	28.59	36.27
CYP2D6 inhibitor (%)	83.88	92.34	91.24	88.36	86.22	89.63	91.19
CYP2D6 substrate * (%)	56.44	56.88	57.39	58.14	55.49	51.70	59.00
CYP3A4 inhibitor (%)	31.48	32.59	33.51	40.29	30.01	33.23	34.31
CYP3A4 substrate * (%)	34.94	33.79	36.14	37.18	35.42	40.46	38.97
**Excretion**							
Half-life * (h)	68.11	67.24	67.41	67.56	67.78	39.82	54.73
HPC * (uL/min/10^6^ cells)	36.10	32.75	36.66	36.18	35.07	45.91	48.32
MSC * (mL/min/g^−1^)	36.05	34.16	35.58	41.62	35.93	30.30	27.97
**Toxicity**							
hERG blockers (%)	35.28	35.97	37.25	37.52	36.65	32.27	32.43
AMES toxicity (%)	43.30	44.46	43.93	42.63	42.30	41.60	40.60
DILI (%)	46.46	42.09	51.68	46.05	50.41	43.00	40.09
ROALD_50_ (mmol/kg)	2.82	2.34	6.92	5.75	4.90	29.51	15.14
hMTD (mg/kg/day)	1.23	1.80	0.70	1.80	0.19	0.19	0.12

Abbreviations: ALO, aloe-emodin; CHR, chrysophanol; EMO, emodin; PHY, physcion; RHE, rhein; DOP, dopamine; LDP, levodopa; TPSA, topological polar surface area; (+), accepted; (−), rejected; C2P, Caco-2 permeability; HIA, human intestinal absorption; BBB, blood–brain barrier; PPBR, plasma protein, binding rate; HPC, hepatocyte clearance; MSC, microsome clearance; hERG, human ether-a-go-go-related gene; DILI, drug-induced liver toxicity; ROALD50, rat oral acute half-maximal lethal dose; hMTD, human maximum tolerated dose. * Parameter models have low confidence prediction [58].

## Data Availability

The data files generated and presented in this study are available upon request of the corresponding author. Prediction of pharmacokinetic properties can be replicated using the webservers.

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
