# Peer review of "In Silico Neuroprotective Effects of Specific Rheum palmatum Metabolites on Parkinson’s Disease Targets"

_ijms, 2023, doi:10.3390/ijms241813929_

Round 1
Reviewer 1 Report
Garcia and colleagues in the present research article entitled ‘Neuroprotection from Rheum palmatum major metabolites on Parkinson’s disease targets in silico’ investigated the potential neuroprotective effects of metabolites derived from Rheum palmatum (Chinese rhubarb) in the context of Parkinson's disease (PD). The main objective of the study was to explore whether certain metabolites from R. palmatum could have therapeutic benefits for PD by targeting specific proteins and molecular pathways associated with the disease. The paper begins by introducing PD as the second most prevalent neurodegenerative disease after Alzheimer's and describes its characteristic motor and non-motor symptoms. Current treatments are limited to managing symptoms due to the lack of a precise understanding of the disease's underlying mechanisms. The authors highlight previous research that identified α-synuclein (ASN) as one of the key proteins involved in PD. ASN overexpression and alternate folding have been linked to the development of intraneuronal aggregates and neurodegeneration. Additionally, standard PD treatments target dopamine regulation through various mechanisms. The study explores natural products as a potential source of neuroprotective compounds, focusing on Rheum palmatum's metabolites. Notably, chrysophanol, rhein, aloe-emodin, emodin, and physcion are abundant in R. palmatum roots. Previous studies have suggested that these metabolites possess redox-mediating activity and neuroprotective properties, particularly emodin and chrysophanol. To investigate the potential neuroprotective effects of these metabolites, the researchers conducted molecular docking simulations with PD-related protein targets. Emodin exhibited the most promising binding activity to the A2A adenosine receptor, comparable to levodopa, a standard PD treatment. Chrysophanol and physcion also showed good inhibitory activity on monoamine oxidase isoform B (MAOB). Furthermore, density functional theory (DFT) analysis was performed to evaluate the molecular properties and stability of the protein-ligand complexes. The metabolites showed favorable stability and binding activity compared to dopamine and levodopa. In silico pharmacokinetic predictions revealed that the metabolites from R. palmatum satisfied Lipinski's rule of five, indicating potential drug-likeness. Although some metabolites showed limited Caco-2 permeability, they exhibited good oral bioavailability and plasma protein binding rates. Overall, the study suggests that metabolites from Rheum palmatum may hold promise as neuroprotective agents in Parkinson's disease. However, further in vitro and in vivo studies are needed to validate their efficacy and safety. The research highlights the potential of natural products as sources of lead compounds for developing new therapeutic strategies against neurodegenerative diseases.
In general, I think the idea of this article is really interesting and the authors’ fascinating observations on this timely topic may be of interest to the readers of International Journal of Molecular Sciences. However, some comments, as well as some crucial evidence that should be included to support the author’s argumentation, needed to be addressed to improve the quality of the manuscript, its adequacy, and its readability prior to the publication in the present form.
Please consider the following comments:
• I recommend revising the title. In its current form, I find it to be relatively long and could be made more concise without losing its essential elements. I would suggest rephrasing or abbreviating certain parts of the title to make it more succinct. A potential revised title could be: "In Silico Neuroprotective Effects of Specific Rheum palmatum Metabolites on Parkinson's Disease Targets." This revision includes the specificity of the metabolites being studied, the research approach (in silico), and the focus on Parkinson's disease targets [1-3].
• A graphical abstract that will visually summarize the main findings of the manuscript is highly recommended.
• Abstract: According to the Journal’s guidelines, this section should be presented as a short summary of about 200 words maximum that objectively represents the article. It should let readers get the gist or essence of the manuscript quickly, prepare the readers to follow the detailed information, analyses, and arguments in the full paper and, most of all, it should help readers remember key points from your paper. Please, consider rewrite this paragraph following these instructions [4].
• Keywords: Please list ten keywords chosen from Medical Subject Headings (MeSH) and use as many as possible in the title and in the first two sentences of the abstract. I would suggest adding “Neuroprotection” and “ADMET properties” as keywords.
• Introduction: The authors need to reorganize this section with several paragraphs made up of about 1000 words, introducing information on the main constructs of this study, which should be understood by a reader in any discipline, and making it persuasive enough to put forward the main purpose of the current research the author has conducted and the specific purpose the author has intended by this protocol. I would like to encourage the authors to present the introduction starting with the general background, proceeding to the specific background on the concept of using natural products as a potential source of lead compounds with pharmacokinetic properties for treating neurological diseases. Those main structures should be organized in a logical and cohesive manner [5].
• In this regard, I believe that the Introduction section would benefit from additional information to enhance its clarity and contextualization. To strengthen this section, I suggest highlighting the understanding of neural substrates involved in Parkinson's disease pathogenesis: for this reason, I would suggest to provide a concise yet informative overview of Parkinson's disease pathogenesis, highlighting the key neural substrates involved in the disease's progression. Mention the role of dopaminergic neurons and the substantia nigra pars compacta (SNpc) in the regulation of motor functions, and their subsequent degeneration leading to the characteristic motor symptoms of Parkinson's disease, such as bradykinesia, resting tremors, and postural instability [6-7]. Furthermore, here Authors can discuss the involvement of dopamine and other neurotransmitters, such as epinephrine and norepinephrine, in Parkinson's disease pathogenesis. Address how the decline in dopamine levels contributes to the motor symptoms observed in patients and the rationale behind targeting dopaminergic pathways for symptomatic treatment [8-10].
• Results: In my opinion, the manuscript does not provide a detailed interpretation of the molecular docking and pharmacokinetic predictions. Authors should thoroughly discuss the significance and implications of the docking results, such as the binding affinities of the metabolites to the protein targets, and how they compare to standard drugs for Parkinson's disease. Also, I believe that the manuscript should compare the predicted inhibitory activities of the R. palmatum metabolites with other known inhibitors of the targeted proteins. This comparative analysis can provide a context for understanding the potential efficacy of the metabolites as neuroprotective agents.
• Material and Methods: I believe that this section would benefit from a clearer structure and better organization of the flow of information. For example, I believe that the section should include information about the specific parameters used for molecular docking, such as grid spacing, docking algorithm, scoring function, and any cutoff values for identifying potential ligand-target interactions. Also, the details of the computational tools or software used for predicting pharmacokinetic properties should be mentioned. Additionally, any specific parameters or thresholds applied during these predictions should be clarified.
• In my opinion, the ‘Conclusions’ paragraph would benefit from some thoughtful as well as in-depth considerations by the authors, because as it stands, it lists down all the main findings of the research, without really stressing the theoretical significance of the study. Authors should make an effort, trying to explain the theoretical implication as well as the translational application of their research.
• In according to the previous comment, I would ask the authors to include a proper and defined ‘Limitations and future directions’ section before the end of the manuscript, in which authors can describe in detail and report all the technical issues brought to the surface,
• References: Authors should consider revising the bibliography, as there are several incorrect citations. Indeed, according to the Journal’s guidelines, they should provide the abbreviated journal name in italics, the year of publication in bold, the volume number in italics for all the references.
I hope that, after these careful revisions, the manuscript can meet the Journal’s high standards for publication. I am available for a new round of revision of this article.
I declare no conflict of interest regarding this manuscript.
Best regards,
Reviewer
References:
1. https://plos.org/resource/how-to-write-a-great-title/
2. https://www.nature.com/nature-index/news-blog/how-to-write-a-good-research-science-academic-paper-title
3. https://www.indeed.com/career-advice/career-development/catchy-title
4. https://www.mdpi.com/journal/ijms/instructions
5. https://dept.writing.wisc.edu/wac/writing-an-introduction-for-a-scientific-paper/
6. DOI: 10.17219/acem/165944
7. https://doi.org/10.3390/ijms24065926
8. DOI: 10.3390/biomedicines11030945
9. https://doi.org/10.3389/fnmol.2023.1217090
10. https://doi.org/10.3390/biomedicines11051248
Minor editing of English language required.
Author Response
Dear Reviewers,
The coauthors and I express our profound gratitude for the comprehensive and constructive comments and suggestions for this manuscript to the reviewers. We believe that such comments drastically improved the quality of our work and were mostly considered in the revised manuscript. While there were suggestions outside the scope of the study, these were deemed valuable for future investigations in line with this study. In the revised manuscript, added and changed texts were typewritten in red font for revision visibility. Please see attached file of our responses to the reviewers’ comments.

Reviewer 2 Report
Dear Authors
The paper explores the potential neuroprotective effects of major metabolites from Rheum palmatum on targets relevant to Parkinson's Disease (PD) using in silico methods. The study delves into an intriguing area with the potential to uncover novel therapeutic interventions for PD.
Comments:
· The paper should provide a detailed description of the in-silico methodologies employed. The types of molecular docking algorithms, scoring functions, and parameters used for docking studies should be clearly mentioned. This would help the readers understand the approach and improve the applicability in predicting interactions between metabolites and PD-related targets.
· In silico studies can provide valuable insights, but they should be validated against experimental data whenever possible. The paper should discuss about available experimental evidence that supports the predicted interactions between Rheum palmatum metabolites and PD-related targets.
· The paper should elaborate more on the rationale for selecting specific PD-related targets for molecular docking. Explaining the role of these targets in PD pathogenesis and their potential relevance to neuroprotection will contextualize the study's focus.
· The interpretation of docking results should be comprehensive. The paper should also consider the practical feasibility, whether the predicted interactions translate into meaningful physiological effects.
· Since In-silico methods are not widely used, discussing the limitations provides a balanced perspective and acknowledges the boundaries of the study.
· The study should have given more emphasis on the discussion part to focus on the potential implications of the findings for drug development and further experimental investigations. This can help highlight how the in-silico predictions might guide future direction.
Conclusion:
The study's focus on exploring the neuroprotective effects of Rheum palmatum metabolites on PD targets through in silico methods is promising. Addressing these comments and concerns will enhance the clarity and relevance of the paper. Once these improvements are made, the paper has the potential to contribute valuable insights to the field of neuroprotection in PD.
Moderate editing of English language required
Author Response

(The authors gave the same response as above.)

Round 2
Reviewer 1 Report
Dear Authors,
I am pleased to acknowledge that you have indeed addressed all of my concerns and queries in a clear and precise manner. Your responses have provided valuable insights into the modifications made to the manuscript in light of my comments. It is evident that you have taken great care to ensure that the revised manuscript aligns more closely with the scientific rigor expected for publication in IJMS.
Upon reviewing the updated version, I find that the inclusion of the additional studies has indeed enriched the understanding of neural substrates associated with Parkinson's disease pathogenesis. The provided studies contribute significantly to the comprehensiveness of the section. However, in order to provide a more holistic view of the complex phenomena underlying neurodegeneration and PD, I believe there's still an opportunity to expand upon certain factors. Specifically, the discussion of he role of dopaminergic neurons and the substantia nigra pars compacta (SNpc) in the regulation of motor functions, and their subsequent degeneration leading to the characteristic motor symptoms of Parkinson's disease, such as bradykinesia, resting tremors, and postural instability could offer a deeper insight into the mechanisms at play (https://doi.org/10.3389/fnmol.2023.1217090; DOI: 10.3390/cells11162607). By incorporating these aspects, the Introduction section would offer a comprehensive overview of the multifaceted processes driving neurodegeneration in PD. This, in turn, would provide readers with a clearer understanding of the intricate nature of PD.
I want to reiterate my appreciation for your responsiveness and willingness to consider these suggestions. I believe that this minor revision will significantly enhance the quality and impact of the Introduction section.
Thank you once again for your dedication to improving the manuscript. I look forward to seeing the continued progress.
Best regards,
Reviewer
Author Response
Thank you very much for your comprehensive comments and suggestions. The pathophysiology of Parkinson’s disease (PD), highlighting the role of the dopaminergic neurons in the SNpc, was briefly introduced as reflected in the revised manuscript. However, to shorten the introduction section, the function and activity of the PD-related protein targets were transferred to the results and discussion section as appropriate. Instead of introducing such information, they were utilized to discuss the implications of the in silico binding activity.
